# Usefulness of Surface Electromyography Complexity Analyses to Assess the Effects of Warm-Up and Stretching during Maximal and Sub-Maximal Hamstring Contractions: A Cross-Over, Randomized, Single-Blind Trial

**DOI:** 10.3390/biology11091337

**Published:** 2022-09-10

**Authors:** Nicolas Babault, Marion Hitier, Carole Cometti

**Affiliations:** 1INSERM UMR1093-CAPS, UFR des Sciences du Sport, Université Bourgogne Franche-Comté, F-21000 Dijon, France; carole.cometti@u-bourgogne.fr; 2Centre d’Expertise de la Performance, UFR des Sciences du Sport, Université Bourgogne Franche-Comté, F-21000 Dijon, France; hitier.marion@orange.fr; 3Chiropractor, Private Practice, F-21000 Dijon, France

**Keywords:** linear analysis, non-linear analysis, detrended fluctuation analysis, entropy, recurrence plot, root mean square, fractals

## Abstract

**Simple Summary:**

Neural alterations following the warm-up phase of a physical activity program are not clearly established with linear-based methods. However, it appears that nonlinear/complexity-based methods are sensitive enough to detect small and subtle changes in the individuals’ neural drive. Accordingly, the present exploratory study aimed to apply different complexity-based methods to surface electromyography (EMG) of hamstring muscles to detect changes in neural activation after a standardized warm-up and after stretching exercises. Maximal and sub-maximal contractions were performed before and after these activities. The resultant EMG signal was processed using a linear analysis (the root mean square) and using nonlinear analyses: sample entropy, the recurrence quantification analysis and the detrended fluctuation analysis. These methods are well known to witness the presence of informationally rich variability. Our results revealed an increase in complexity of the EMG signal after warm-up and stretching during both maximal and sub-maximal contractions. In contrast, the linear analysis did not show any alteration. We concluded that, as for neuromuscular fatigue, complexity-based methods are sensitive enough to detect subtle changes in the EMG signal. The so-observed increase in complexity after warm-up and stretching would suggest that the neuromuscular system is in an optimized state for subsequent neuromuscular activity.

**Abstract:**

This study aimed to apply different complexity-based methods to surface electromyography (EMG) in order to detect neuromuscular changes after realistic warm-up procedures that included stretching exercises. Sixteen volunteers conducted two experimental sessions. They were tested before, after a standardized warm-up, and after a stretching exercise (static or neuromuscular nerve gliding technique). Tests included measurements of the knee flexion torque and EMG of biceps femoris (BF) and semitendinosus (ST) muscles. EMG was analyzed using the root mean square (RMS), sample entropy (SampEn), percentage of recurrence and determinism following a recurrence quantification analysis (%Rec and %Det) and *a* scaling parameter from a detrended fluctuation analysis. Torque was significantly greater after warm-up as compared to baseline and after stretching. RMS was not affected by the experimental procedure. In contrast, SampEn was significantly greater after warm-up and stretching as compared to baseline values. %Rec was not modified but %Det for BF muscle was significantly greater after stretching as compared to baseline. The *a* scaling parameter was significantly lower after warm-up as compared to baseline for ST muscle. From the present results, complexity-based methods applied to the EMG give additional information than linear-based methods. They appeared sensitive to detect EMG complexity increases following warm-up.

## 1. Introduction

Most physical activities or sport practices start with a multi-component preparatory activity commonly called warm-up. Warm-up includes various exercises at low and high-intensity aiming to improve performance and prevent injuries [1,2]. Part of the beneficial effects of warm-up relies on temperature-dependent mechanisms such as a decreased viscous resistance or nerve conduction rate [3]). Other mechanisms, not temperature-dependent, such as those implied in post-activation potentiation [4], may acutely affect muscle contractility.

Amongst the various activities conducted during warm-up, stretching exercises are very popular [5]. They are usually applied to increase joint range of motion, muscle performance or reduce activity-related injury risks [5]. These expected effects may have distinct efficiencies depending on the stretching technique used [6]. For this reason, authors recommended using dynamic stretching rather than static stretching in warm-up routines [7,8]. Indeed, starting from the late 1990s [9], authors have concluded static stretching transiently decreased muscle strength. However, stretching duration, modality, intensity or the inclusion of stretching inside dynamic activities are well-described key factors that significantly impact strength or more generally neuromuscular alterations [10,11,12,13,14].

The immediate effects of stretching modalities on the neuromuscular system (and more particularly of static stretching) are frequently investigated in the literature. Most techniques target the muscle–tendon complex and less is known on techniques focusing on the neural tissue. For instance, the neurodynamic nerve gliding is a technique used in clinical practices that aims to restore the dynamic balance between the relative movement of neural tissues and the surrounding mechanical interfaces [15]. In a recent study, authors have demonstrated that the neurodynamic nerve gliding technique had slightly greater effects on hamstring stiffness than static stretching in physically active men [16]. Similar or greater increases in range of motion have been observed with acute or chronic use of nerve-oriented stretching (such as neurodynamic nerve gliding) as compared to the other techniques [17,18,19,20]. The larger effects were likely due to a decreased nerve tension and increased strain in connective tissues [16]. However, to the best of our knowledge, no study has explored whether neurodynamic stretching could be included in warm-up routines.

While the immediate mechanical effects of stretching, are often explored, less is known for neural alterations and conclusions remain equivocal. Depending on the stretching characteristics, muscle activation could decrease after stretching [21,22] or remain unaltered [14,23]. Warm-up effects on muscle activation also remain ambiguous. For instance, most studies obtained unaltered muscle activation [24,25]. However, neural alterations could not be excluded since authors suggested some neural modulations during sub-maximal contractions [26].

Muscle activation is generally investigated by using surface electromyographic (EMG) activity, an interference signal increasing with the contraction intensity. A time-domain analysis is generally performed by using the root mean square value. However, EMG appeared to be non-linear and could be explored using complexity-based methods that witness the presence of informationally rich variability, independently of the amplitude of its fluctuation [27,28,29]. High complexity is known to characterize healthy physiological state while a loss of complexity corresponds to a sub-optimal state. These analyses are expected to be more sensitive to classic linear methods and could give additional information. For instance, in a recent review, authors indicated that some complexity analyses could be related to motor unit firing rates or may be efficient to reveal early onset of muscle fatigue [28]. A loss of complexity is generally obtained with muscle fatigue or with high-speed locomotion [30,31]. However, to the best of our knowledge, no study has attempted to determine whether complex analyses could be sensitive enough to detect neuromuscular changes after warm-up and stretching activities.

The aim of the present study was to use different complexity-based methods applied on the EMG signal to determine their sensitivity to detect neuromuscular changes following realistic warm-up procedures generally used in athletes that included some stretching exercises. As for fatigue, we hypothesized complexity-based analyses would be sensitive to differentiate the physiological state of the neuromuscular system following warm-up and stretching. An increase in complexity was hypothesized following warm-up which could suggest an optimized neuromuscular state for subsequent performance. A special interest was given to explore passive and neurodynamic nerve gliding stretching techniques following a similar comprehensive warm-up.

## 2. Materials and Methods

### 2.1. Study Design

This study was a cross-over, randomized and single-blind trial. All volunteers came to the laboratory on three separate occasions (familiarization, and two experimental sessions) with seven days between each session. All sessions were performed the same day of the week, at the same hour of the day. During the total duration of the study, volunteers were instructed to continue their usual training habits but to refrain from intensive exercise at least during the two days preceding the experimental testing sessions.

The familiarization session aimed to (i) explain the experimental procedure, (ii) determine anthropometrics (age, height and body mass), and (iii) familiarize volunteers with the different tests and warm-up exercises. During the two experimental sessions (Figure 1), data were collected at three different time points: (i) before, (ii) immediately after a standardized warm-up, and (iii) immediately after a stretching exercise. Stretching was applied using either the static modality or neurodynamic nerve gliding technique. Both stretching modalities were time-matched and randomly presented using www.randomizer.org (accessed on 2 February 2022) website. Limb dominance was ignored, and tests were conducted for the right knee flexors on an isokinetic dynamometer (Biodex 4 Quickset, Biodex Corporation, Shirley, NY, USA). Tests included maximal and sub-maximal knee flexions in isometric conditions. Two maximal voluntary contractions were performed first and followed by a single sub-maximal voluntary contraction. Torque and EMG activity from the long head of biceps femoris (BF) and semitendinosus (ST) muscles were recorded.

### 2.2. Participants

Sixteen volunteers (10 men and 6 women) were included for this study. Their mean age ± standard deviation (SD), height, and body mass were 21.3 ± 2.0 years, 172.3 ± 10.2 cm, and 64.3 ± 11.3 kg, respectively. All were students from the sport science faculty in Dijon and all were competitive athletes from track and field, handball and soccer (*n* = 8, 4, and 4, respectively) with an average of 7.7 ± 3.6 h training per week. None reported lower limb injuries or back pain within the last three months or specific injuries located in hamstring muscles during the past two years (exclusion criteria). Prior to participation, they were fully informed about the purpose of the study and experimental procedure. All signed an informed consent form. This study was conducted according to the declaration of Helsinki. Approval was obtained from the Ethics Committee for sport science research (CERSTAPS; IRB00012476-2022-15-03-166). The sample size was calculated a priori using G*Power (version 3.1.9.6, free software available at https://www.psychologie.hhu.de/arbeitsgruppen/allgemeine-psychologie-und-arbeitspsychologie/gpower.html (accessed on 10 October 2019)) with the following criteria: effect size of 0.35, power of 0.8, probability error of 0.05 with the main outcome being the root mean square. A sample size of 16 individuals was indicated.

### 2.3. Experimental Sessions

During the two experimental sessions (Figure 1), volunteers had to perform a standardized warm-up [13]. Warm-up started with 10 min cycling using an indoor ergocyle (CyclOps 400 Pro equipped with PowerTap, Madison, WI, USA) with saddle and handlebar settings individually adjusted. A constant power output was chosen (80 W and 100 W for women and men, respectively). Volunteers then performed submaximal contractions of the lower limb muscles: 6 concentric knee extensions, 12 lunges, 10 concentric plantar flexions, 10 eccentric plantar flexions, 10 concentric knee flexions and 10 eccentric knee flexions. These submaximal contractions were followed but athletic drills: 20 rapid high knees and 20 rapid butt kicks. Finally, volunteers performed five maximal jumps (counter movement jump style using arms).

After this warm-up, volunteers conducted a stretching exercise: either static or neurodynamic nerve gliding according to randomization. Static stretching (6 × 30 s) was performed while lying supine on a massage table. It consisted in a passive straight leg rise on the right side. Briefly, the experimenter placed a hand on the right knee and the other hand on the ipsilateral foot. Hands aimed to maintain the knee extended and the ankle in dorsiflexion. Then, the straight leg was slowly and progressively raised until the discomfort endpoint (but not pain). The position was maintained for 30 s. The lower limb was then positioned back on the table for 15 s. The passive static stretching was performed six times. Neurodynamic nerve gliding technique, a type of neural mobilization, used almost the same configuration. Volunteers were lying supine on a massage table with a slightly flexed cervical and thoracolumbar spine supported by a cushion as previously described [16]. The experimenter placed a hand to maintain the right knee extended and the other hand on the ipsilateral foot. The experimenter slowly and progressively raised the straight leg until the discomfort endpoint (but not pain). Then, the volunteer performed cervical extension concomitantly with dorsiflexion. Subsequently, cervical flexion was performed again with concomitant plantarflexion. This cycle was repeated for 30 s. The lower limb was then positioned back on the table for 15 s. The neurodynamic nerve gliding stretching was performed six times.

Tests were conducted at three time points: before warm-up (called BASELINE), immediately after the warm-up (called WARM) and immediately after stretching procedure (called STRETCH). Volunteers were seated on the Biodex dynamometer has previously described [32]. Briefly, the left side was maintained with an approximately 100° hip angle and the right hip was flexed with a 45° angle between the thigh and torso. Then, the axis of the right knee joint (corresponding to the lateral condyle) was aligned with the input axis of the dynamometer according to the manufacturer’s guidelines. A dynamometer strap was firmly attached distally on the subject’s right thigh and a second one above the subject’s right ankle. Velcro straps were also applied over the pelvis, trunk and contralateral thigh to minimize inappropriate displacements.

Tests included two maximal voluntary isometric contractions of the hamstring muscles (i.e., knee flexion). Volunteers were requested to contract as maximally as they could for 5 s. Rest between the two maximal voluntary contractions was 15 s long. Then, volunteers were requested to conduct a 5 s sub-maximal voluntary isometric contraction. This contraction was performed at 40% of the maximal voluntary torque 15 s after the last maximal voluntary contraction. Volunteers visualized the contraction intensity on the dynamometer screen (predetermined horizontal line). All voluntary contractions were performed with a 90° knee flexion angle.

### 2.4. Data Recording, Preprocessing and Initial Analyses

Torque and EMG activity of the long head of BF and ST muscles were recorded using a Biopac MP150 system and associated software (AcqKnowledge 4.2 for MP systems, Biopac System, Santa Barbara, CA, USA) at a 1000 Hz sampling frequency. EMG used two pairs of silver chloride surface electrodes placed over muscle bellies according to SENIAM recommendations. The interelectrode distance was 2 cm (center to center). The reference electrode was fixed to the right patella. Low impedance (<5 kW) of the skin-electrode interface was obtained by shaving, abrading with sandpaper and cleansing with alcohol. The EMG signals were amplified with a bandwidth frequency ranging from 10 to 500 Hz (common mode rejection ratio = 110 dB, gain = 500).

Analyses of the maximal voluntary contractions started with a manual identification of the maximal voluntary torque value and timepoint using AcqKnowledge software (AcqKnowledge 4.2 for MP systems, Biopac System, Santa Barbara, CA, USA). The best contraction, as witnessed by the maximal voluntary torque, was only retained for subsequent analyses. Then, torque and raw EMG signals were extracted over a 1 s time window being 0.5 s apart the so-identified maximal voluntary torque. Torque and EMG were also extracted using 1 s windows within the sub-maximal voluntary contraction plateau (i.e., between seconds 3 and 4). EMG raw data were band-pass filtered (20–300 Hz, 4th order) and root mean square (RMS) was calculated over 250 ms sliding windows using the “biosignalEMG” R package [33]. The maximal voluntary torque and corresponding maximal RMS values were calculated. The mean torque and RMS values were calculated during sub-maximal voluntary contractions. An example of the EMG signal processing during maximal and submaximal contraction is shown in Figure 2. Values were calculated using a custom-written R routine. Complex analyses were subsequently conducted and included various tools: Sample Entropy (SampEn), Recurrence Quantification Analysis (RQA) and Detrended Fluctuation Analysis (DFA).

### 2.5. Sample Entropy

SampEn is often used as index of physiological complexity [28,34]. For instance, complexity (and SampEn) has been shown to be greater in concentric contractions and to decrease with fatigue [30]. SampEn was calculated using the “TSEntropies” R package [35]. To be computed, EMG time series *x_i_* was first embedded in a delayed *m*-dimensional space. The probability *B^m^*(*r*) that two sequences match for *m* points was computed by counting the average number of vector pairs within a tolerance of *r*. The embedding dimension *m* was set to length 2 and tolerance *r* to 0.2 multiplied by the SD of the original EMG time series [30]. This process was repeated for an embedding dimension of *m* + 1 to obtain the probability *A^m^*(*r*). SampEn was the negative logarithm of the ratio between these two probabilities:
(1)
$$SampEn\;\left(m,r\right)=-ln\left(A^{m}\left(r\right)/B^{m}\left(r\right)\right)$$


### 2.6. Recurrence Quantification Analysis

RQA is used to detect recurring patterns and non-stationarities in a dynamic system [28,36]. Detailed descriptions could be found in previous studies [28,37,38,39]. A time delay *t* was used to embed the EMG time series into a *d*-dimensional space. Thus, the first step of RQA was to construct the time delay vectors *y_i_* by cutting the original EMG time series *x_i_* into vectors:
(2)
$$y_{i}\left(d\right)=\left(x_{i},\;x_{i+\tau },\;x_{i+\left(d-1\right)\tau }\right)$$


This method requires to determine the time delay *t* and the minimum embedding dimension *d*. These parameters were first optimized according to Marwan et al. [40]. Then, the recurrence plot was built. It consists in a two-dimensional plot that computes the distances between the delayed vectors *y_i_*(*d*) and *y_j_*(*d*). Distances are normalized according to the maximum distance. A recurrence occurs when the distance between *y_i_*(*d*) and *y_j_*(*d*) is smaller than a threshold radius *r*. The recurrence is shown with a black dot.

Authors [41] indicated that fine changes in surface EMG can be detected by the percentage of recurrence (%Rec) and percentage of determinism (%Det). %Rec quantifies the density of the recurrent points and reflects the current state of the system. %Det quantifies the ratio of the recurrence points that form diagonals higher than a minimal length (*l_min_*) and reflects the amount of rule-obeying structure in the signal dynamic. A deterministic system produces longer recurrence diagonals and fewer isolated points than uncorrelated random time series.

RQA and parameters’ optimization were achieved using “crqa” R package [42]. In the present study, the *l_min_* was set at 3 points, the radius *r* was fixed at 25% of the maximum distance to limit too high %Rec [37]. The time delay *t* was set at 5 and the embedding dimension *d* was 10. An example of recurrence plots is shown in Figure 3.

### 2.7. Detrended Fluctuation Analysis

DFA, a complexity method, is used to assess the self-similarity of a signal [43]. The algorithm gives a scaling parameter *a* characterizing a correlation feature (related to the Hurst exponent). To determine *a*, the cumulative sum *z_i_* of the EMG time series *x_i_* was first calculated. Then, *z_i_* was divided into non-overlapped boxes of equal size *n*. The local trend *z_i_*(*n*) was determined in each box by using a least-squared linear detrending and the difference with *z_i_* was computed. The average fluctuation *F*(*n*) for a given box length was calculated as the root-mean-square of the variance of the residuals:
(3)
$$F\left(n\right)=\sqrt{\frac{1}{N}\overset{N}{\underset{k=1}{\sum }}{\left[z_{i}-z_{i}\left(n\right)\right]}^{2}}$$


These steps were repeated for different box sizes *n*. The scaling parameter *a* was calculated as the slope of the log-log fitting *F*(*n*) against *n* (Figure 3). *a* < 0.5 indicates an anti-correlation between samples. If 0.5 < *a* < 1, the signal displays long-range correlations, characteristic of fractal dynamics. If *a* > 1, correlations exist but not in the form of persistent power-law correlations. Finally, *a* = 0.5, *a* = 1 and *a* = 1.5 correspond to white noise, pink noise and brown noise, respectively. The scaling exponent *a* has previously been shown to increase with fatigue which corresponds to a loss of complexity [30]. Box length for DFA was set at 4 and DFA was calculated across time scales (40 boxes ranging from 4 to 251 data points). DFA was conducted using “DFA” R package [44].

### 2.8. Statistical Analyses

Statistical analyses were conducted using JASP (version 0.14, JASP Team 2020, University of Amsterdam, available free at https://jasp-stats.org/download/ (accessed on 9 September 2021)). The normality and sphericity of the data were tested and confirmed by the Shapiro–Wilk and Mauchly’s tests. Then, a two-way (condition × time) analysis of variances (ANOVA) with repeated measures was performed. The “condition” factor corresponded to the two experimental conditions. The “time” factor corresponded to the three tests (BASELINE vs. WARM vs. STRETCH). In case of non-sphericity, a Greenhouse–Geisser correction was applied. Post hoc tests with Bonferroni corrections were conducted if significant main effects or interactions were present. Partial eta square (partial *η^2^*) was calculated from ANOVA results, with values of 0.01, 0.06 and above 0.14 representing small, medium and large differences, respectively [45]. Subsequently, qualitative descriptors of standardized effects were used for pairwise comparisons with Cohen’s d < 0.5, 0.5–1.2 and > 1.2 representing small, medium and large magnitudes of change, respectively [45]. *p* < 0.05 was taken as the level of statistical significance for all comparisons. Absolute values are expressed as mean ± SD or mean difference with 95% confidence intervals (95%CI). 

## 3. Results

Maximal voluntary torque was significantly altered depending on time (Table 1). Torque was significantly greater during WARM as compared to BASELINE (mean difference (95%CI): 5.0 (0.7;9.3); d = 0.745; medium; *p* = 0.017) and STRETCH (mean difference (95%CI): 8.2 (3.9;12.5); d = 1.211; large; *p* < 0.001) (Figure 4). No difference was obtained between BASELINE and STRETCH (mean difference (95%CI): 3.1 (−1.1;7.4); d = 0.466; small; *p* = 0.216). BF RMS was not modified during the experimental procedure (Table 1). In contrast ST RMS demonstrated significant main effects for condition and time (Table 1). Post hoc analyses revealed greater ST RMS during the static as compared to the neurodynamic nerve gliding condition (mean difference (95%CI): 0.055 (0.000;0.110); d = 0.539; medium; *p* = 0.048). Considering the time effect, no significant difference was observed during post hoc analyses instead of the significant main ANOVA effect. Finally, during sub-maximal voluntary contractions, no difference was obtained for torque, BF RMS and ST RMS (Table 1).

SampEn demonstrated a significant time effect during maximal contractions for both BF and ST muscles (Table 1). Post hoc tests revealed that SampEn for BF muscle was significantly greater after WARM (mean difference (95%CI): 0.101 (0.040;0.163); d = 1.042; medium; *p* < 0.001) and STRETCH (mean difference (95%CI): 0.090 (0.029;0.152); d = 0.928; medium; *p* = 0.002) as compared to BASELINE (Figure 4). No difference was obtained between WARM and STRETCH (mean difference (95%CI): 0.011 (−0.073;0.051); d = 0.114; small; *p* = 1.0). Similarly, SampEn for ST muscle was significantly greater after WARM (mean difference (95%CI): 0.105 (0.053;0.158); d = 1.268; large; *p* < 0.001) and STRETCH (mean difference (95%CI): 0.116 (0.064;0.169); d = 1.403; large; *p* < 0.001) as compared to BASELINE without any difference between WARM and STRETCH (mean difference (95%CI): 0.011 (−0.041;0.064); d = 0.135; small; *p* = 1.0). No SampEn difference was obtained during sub-maximal contractions.

RQA was analyzed using %Rec and %Det. No difference was observed for %Rec (Table 1). %Det only revealed a significant time effect for BF during maximal contractions. More particularly, %Det for BF was significantly greater after STRETCH as compared to BASELINE (mean difference (95%CI): 3.48 (0.05;6.42); d = 0.753; medium; *p* = 0.016) (Figure 4). No difference was observed for WARM as compared to BASELINE (mean difference (95%CI): 2.20 (−0.73;5.13); d = 0.476; small; *p* = 0.200) and STRETCH (mean difference (95%CI): 1.28 (−1.65;4.22); d = 0.277; small; *p* = 0.830).

The *a* scaling parameter was not affected by the experimental procedures for BF muscle. In contrast, a significant time effect was observed for ST during maximal and sub-maximal contractions (Table 1). Post hoc analyses conducted on the main time effect for ST revealed lower *a* during WARM maximal contractions as compared to BASELINE (mean difference (95%CI): 0.042 (0.015;0.069); d = 0.975; medium; *p* = 0.002) (Figure 4). No difference was obtained between STRETCH and WARM (mean difference (95%CI): 0.023 (−0.005;0.050); d = 0.526; medium; *p* = 0.132) and BASELINE (mean difference (95%CI): 0.019 (−0.008;0.046); d = 0.449; small; *p* = 0.247) time points. During sub-maximal contractions, *a* values for ST during WARM (mean difference (95%CI): 0.046 (0.021;0.070); d = 1.162; large; *p* < 0.001) and STRETCH (mean difference (95%CI): 0.032 (0.007;0.057); d = 0.810; medium; *p* = 0.009) were significantly lower than BASELINE without any difference between WARM and STRETCH (mean difference (95%CI): 0.014 (−0.011;0.039); d = 0.352; small; *p* = 0.509).

## 4. Discussion

The present study attempted to apply non-linear/complexity-based methods to EMG signals following a standardized warm-up that included stretching exercises. Different methods including SampEn, RQA and DFA were used. Our results revealed complexity-based methods can be used to detect changes in the physiological state of the neuromuscular system following warm-up and stretching. While the standard linear analysis (i.e., RMS) was unable to detect any changes, complexity-based methods confirmed our a priori hypothesis with some significative increases in the EMG signal complexity. This increased complexity would suggest subtle changes following warm-up that could optimize the neuromuscular system for subsequent neuromuscular performance.

Warm-up is a preparatory activity with multiple beneficial consequences for individuals such as performance improvements for competitive athletes. The present study confirmed this most-desired effects since knee flexion torque was significantly increased after the warm-up. Such result is often obtained in the literature with various conditioning exercises on single joint or whole-body outcomes [1,46,47]. In contrast, our results indicated maximal torque returned to baseline after stretching whatever the modality applied (static or neurodynamic nerve gliding technique). This finding partly confirmed the literature since stretching has often been shown to be detrimental for subsequent performance [7,10,13,48]. While the acute effects of static stretching on strength are often documented in the literature, less is known for neurodynamic nerve gliding. This method has previously been shown to produce larger gains in range of motion as compared to other techniques [17,18,19,20]. However, no study has explored the immediate effect on the subsequent force. According to our results, we can conclude that this technique is as detrimental as static stretching for force production when included at the end of a comprehensive warm-up.

The torque outcomes are undoubtedly in accordance with previously published observations. However, neural alterations after warm-up or stretching remain equivocal. Following warm-up, a lack of alteration in muscle activation is often registered [24,25,49]. However, some alterations are suggested such as increases in muscle conduction time [50,51], spinal excitability [52] or changes during sub-maximal contractions [26]. Following stretching, changes in spinal or cortical excitability have been observed [14,53] without being associated with clear changes in peak EMG activity [14,23,54]. In the present study, and as previously registered, the RMS amplitude of the EMG signal was not modified during and following our two experimental conditions. The lack of changes could be due to the specificity of the acute alterations (i.e., depending on the exercises performed or outcomes) but could also be attributed to the low resolution of the EMG technique to detect small activation changes [55]. Accordingly, the present study gives new insight for neural changes following warm-up and stretching. Non-linear analyses would show subtle changes in the EMG signal that might have large consequences for subsequent activities. 

In the present study, different non-linear approaches were applied to the EMG signal. First, it is important to note that non-linear methods are regularly used on biological signals (e.g., force output, EMG, electrocardiography, electroencephalography) to explore fatigue-related mechanisms, to detect various movement patterns and to detect diseases [28,30,37,39,43,56,57,58]. To the best of our knowledge, the present study is the first that applied these methods to the EMG signal to explore the effects of warm-up and stretching exercises in athletes. Entropy was previously used following stretching. However, studies mostly considered patients and used high-density surface EMG [59].

The analyses applied here revealed small changes during both experimental procedures (no difference between conditions). Taken as a whole, an increase in complexity is observed after warming up. It is witnessed by an increase in entropy for both hamstring muscles, a decrease in *a* DFA (only for ST muscle) and a slight increase in %Det for BF muscle after stretching. The increase in complexity refers to less irregularities in the EMG signal with less random information [60]. The slight but significant *a* DFA decrease during maximal and sub-maximal contractions partly illustrated this behavior with values going closer to 1 (characteristics of an optimized system) and farther to 1.5 (characteristics of the brown noise).

Alterations of complexity have been linked to some physiological mechanisms. For instance, authors attributed changes in entropy to changes in action potential amplitude and velocity [28]. In addition, the *a* DFA has been suggested to characterize the level of motor unit recruitment [30] and %Det to motor unit synchronization, depolarization of the sarcolemma or conduction velocity [37,38]. Similar associations could be made here such as an increase in conduction velocity, already registered following warm-up [50,51], that could have been witnessed by the increase in entropy [28]. However, such causal links would be highly speculative since no direct measurement of these different neural recruitment characteristics have been conducted here. Similarly, numerous factors and mechanisms of regulation have not yet been questioned with complexity-based methods and require further investigations (e.g., excitability).

Notwithstanding, these preliminary findings are functionally relevant for individuals included in a physical activity program (whatever it is for sport, health or physiotherapy). Indeed, because complexity is recognized as defining a healthy physiological state, the increase of complexity, observed here, depicted an optimized neuromuscular system for subsequent activity. Functionally speaking, individuals would be in an optimum state to perform contractions (maximal as well as sub-maximal). In addition, it can be speculated individuals would be in an optimum state for short- or long-term adaptive capacity such as for motor learning (e.g., sport-specific technical skills for athletes or rehabilitation programs based on balance for patients). For instance, authors have previously shown that older adults with a higher complexity during balance exhibited greater adaptive capacities during a specific time to failure balance test [61]. It can be suggested that specific exercises (either voluntarily performed or virtually performed using motor imagery) should therefore be included in a “complete” warm-up to exacerbate the complexity-induced effects of the warm-up. Interestingly, complexity increase is maintained following stretching for SampEn, *a* DFA during sub-maximal contraction and is only obtained after stretching for %Det. Although (static) stretching is not recommended within a warm-up session [7] and apart from a lower force output after stretching as compared to post-warm-up, our results revealed that the beneficial effects of warm-up on complexity are not vanished with subsequent stretching. After stretching, individuals would seem to stay in an optimized neuromuscular state for subsequent activity.

It is important to note that some complex analyses, but not all, revealed alterations during and after our experimental procedures. From the present results, it appeared entropy and DFA are sensitive enough to detect neuromuscular changes after warming up. In contrast, the sensitivity of RQA (%Rec and %Det) was lower. Such conclusion is partly in accordance with the literature. For instance, authors concluded %Rec resulted in a poor correlation with spectral EMG variables [38]. Moreover, RQA has been shown to be sensitive to fatigue but not to the opposed potentiation mechanism [37], a mechanism that could also be achieved during warm-up activities. Moreover, other authors concluded RQA was less effective than entropy to detect fatigue [62]. Our results reinforce this conclusion following warm-up. Such low sensitivity could also be attributed to some methodological considerations such as the surface EMG recording process or more specifically to the parameters implemented in the RQA routine (time delay and embedding dimension). In the present study, the minimal parameters were first determined before performing the RQA. The so-determined parameters were in general accordance with previous studies [28]. SampEn and DFA also applied previously used parameters [30]. Accordingly, our values are in the same range than these studies. The length of the time series might also impact our results. In the present study, 1 s windows were used like in previous experiments [30] but longer time windows could be of interest to increase the methods’ sensitivity. In addition, multiscale entropy or multifractal DFA, could have been conducted. However, the larger sensitivity of these complementary methods is not yet clearly demonstrated [62]. Finally, it is important to acknowledge that athletes from various sport, competitive level and training/warm-up history and experience have been considered here. This heterogeneity might have impacted our results since warm-up exercises and intensity (e.g., pedaling power output) were standardized.

## 5. Conclusions

The present work is the first that explored the effects of warm-up that included stretching exercises with some complexity-based methods (SampEn, RQA, DFA) applied to the surface EMG signal. It is concluded that these methods are sensitive to detect some neuromuscular alterations (increase in complexity) following a comprehensive and realistic warm-up. However, all methods are not as sensitive to detect these subtle changes. RQA appeared less sensitive than SampEn. While the linear amplitude-based EMG analysis (RMS) was not modified by the experimental procedure, non-linear methods revealed increased complexity. The complexity increase obtained following warm-up suggested an optimized neuromuscular system that might be more efficient for subsequent physical activities. Non-linear complexity-based analyses therefore appeared efficient to give additional information than linear EMG analyses.

## Figures and Tables

**Figure 1 biology-11-01337-f001:**
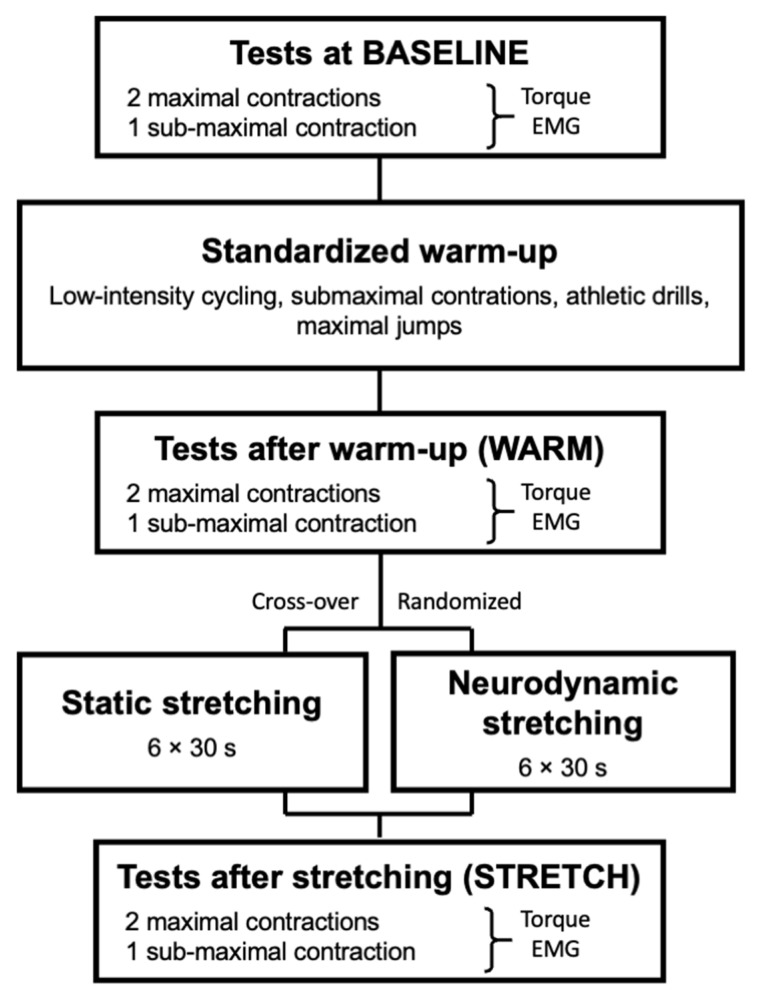
Overview of the two experimental sessions.

**Figure 2 biology-11-01337-f002:**
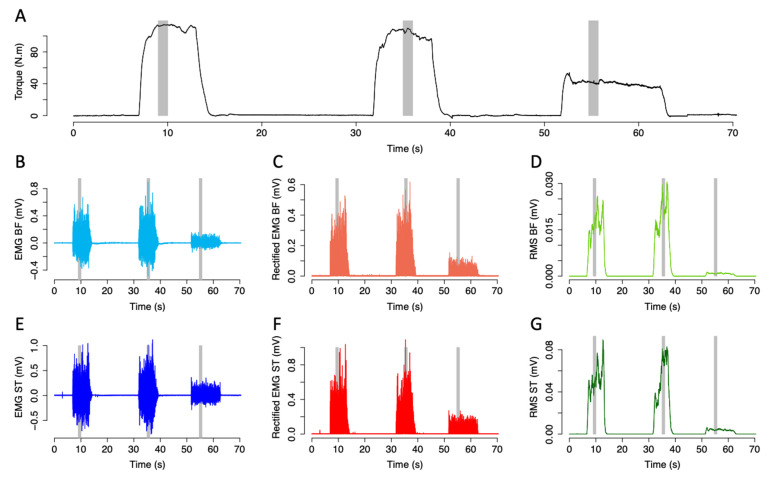
Torque and electromyography (EMG) traces for biceps femoris (**B**,**F**) and semitendinosus (ST) during maximal and sub-maximal contractions for a representative volunteer. Torque (**A**), raw EMG (**B**,**E**), filtered and rectified (EMG)(**C**,**F**) and root mean square (RMS; (**D**,**G**)). During maximal contractions and sub-maximal contractions, 1 s windows were considered for analyses (gray area).

**Figure 3 biology-11-01337-f003:**
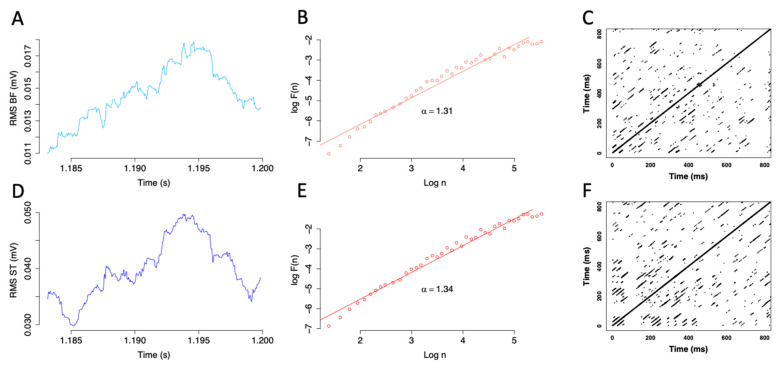
Electromyography (EMG) linear and non-linear analyses for biceps femoris ((**B**,**F**), upper panel) and semitendinosus (ST, lower panel) during a maximal contraction for a representative volunteer. Sliding root mean square (RMS; (**A**,**D**)), detrended fluctuation analysis (**D**,**F**,**A**) with the associated *a* scaling parameter (**B**,**E**), and recurrence quantification analysis (RQA; (**C**,**F**)).

**Figure 4 biology-11-01337-f004:**
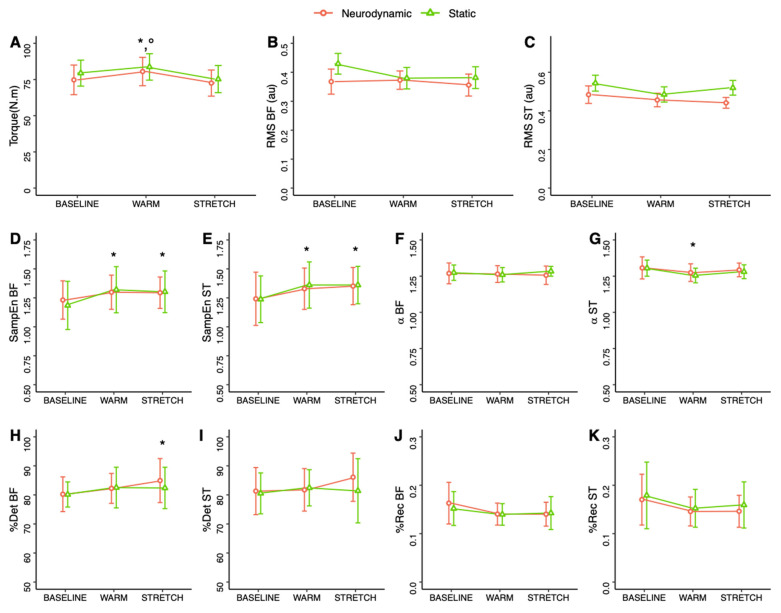
Torque (**A**), root mean square (RMS; (**B**,**C**)), sample entropy (SampEn; (**D**,**E**), *a* scaling parameter (**F**,**G**), percentage of recurrence and determinism (%Rec and %Det; (**H**–**K**)) for biceps femoris and semitendinosus muscles (BF and ST, respectively) during maximal voluntary contractions. Values are mean ± standard deviation. *: significant difference with respect to BASELINE, °: significant difference with STRETCH (*p* < 0.05).

**Table 1 biology-11-01337-t001:** Results of the two-way repeated measures ANOVA.

Variable	Effect	Maximal Contraction			Sub-Maximal Contraction		
		F	*p*	ph2	F	*p*	ph2
	Condition	1.363	0.261	0.083	0.620	0.443	0.040
Torque	Time	11.942	< 0.001 *	0.443	2.443	0.104	0.140
	Condition x Time	0.268	0.767	0.018	0.548	0.532	0.035
	Condition	1.345	0.264	0.082	1.194	0.292	0.074
RMS BF	Time	0.956	0.396	0.060	2.352	0.113	0.136
	Condition x Time	0.427	0.656	0.028	1.155	0.329	0.071
	Condition	4.650	0.048 *	0.237	0.722	0.409	0.046
RMS ST	Time	3.320	0.050 *	0.181	2.258	0.122	0.131
	Condition x Time	0.291	0.750	0.019	1.866	0.172	0.111
	Condition	0.031	0.863	0.002	3.865	0.068	0.205
SampEn BF	Time	10.463	< 0.001 *	0.411	0.097	0.907	0.006
	Condition x Time	1.794	0.184	0.107	0.854	0.436	0.054
	Condition	0.136	0.717	0.009	1.812	0.198	0.108
SampEn ST	Time	19.175	< 0.001 *	0.561	0.796	0.461	0.050
	Condition x Time	0.559	0.578	0.036	0.909	0.414	0.057
	Condition	1.041	0.325	0.069	0.994	0.335	0.062
%Rec BF	Time	0.952	0.346	0.064	0.930	0.350	0.058
	Condition x Time	0.965	0.343	0.064	1.046	0.323	0.065
	Condition	0.881	0.364	0.059	0.225	0.642	0.015
%Rec ST	Time	0.944	0.348	0.063	0.605	0.482	0.039
	Condition x Time	0.966	0.343	0.065	1.453	0.251	0.088
	Condition	0.837	0.375	0.053	0.416	0.529	0.027
%Det BF	Time	4.638	0.018 *	0.236	2.259	0.122	0.131
	Condition x Time	0.549	0.583	0.035	1.954	0.159	0.115
	Condition	1.496	0.240	0.091	2.098	0.168	0.123
%Det ST	Time	1.101	0.330	0.068	1.949	0.160	0.115
	Condition x Time	1.528	0.233	0.092	0.027	0.973	0.002
	Condition	0.557	0.467	0.036	0.096	0.761	0.006
*a* BF	Time	0.285	0.754	0.019	0.232	0.794	0.015
	Condition x Time	1.378	0.268	0.084	3.642	0.055	0.195
	Condition	0.709	0.413	0.045	0.013	0.911	0.001
*a* ST	Time	7.623	0.002 *	0.337	11.367	< 0.001 *	0.431
	Condition x Time	0.394	0.615	0.026	1.118	0.340	0.069

RMS: root mean square, SampEn: sample entropy; %Rec: % recurrence; %Det: % determinism; BF: biceps femoris; ST: Semitendinosus. *: *p* < 0.05.

## Data Availability

Data are freely available at http://dx.doi.org/10.17632/pvf54wjg8y.1 (accessed on 2 August 2022).

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
