# Peer review of "Usefulness of Surface Electromyography Complexity Analyses to Assess the Effects of Warm-Up and Stretching during Maximal and Sub-Maximal Hamstring Contractions: A Cross-Over, Randomized, Single-Blind Trial"

_biology, 2022, doi:10.3390/biology11091337_

Round 1
Reviewer 1 Report
General comment
Thank you for submitting your manuscript. Your study is interesting in using new analytical methods. However, many methodological problems cannot be ignored. In addition, the explanation of clinical significance about the adopted analysis method is not enough. We hope that the following points will be taken into consideration and that this research will be further developed.
Title
From this title, the purpose and design of the study could not be inferred. I think it would be better to use a different expression such as "Investigating the usefulness (or sensitivity) of complexity analysis...” to better link to the purpose of this study. It should also state in the title that this is a crossover design.
Page3, Line 105
I have difficulty understanding from the text that follows that this was a crossover design. It is not clear whether the data were measured three times, including familiarization, or whether the data were measured only during the two-day experimental session. Also, you should illustrate or clearly describe what conditions are present.
Page 3, lines 120-121.
The subject's leg was listed as the right leg, but I think you need to state why it is the right leg. Also, the subject's dominant leg was not stated, and I think it is important to state the dominant and non-dominant leg.
Page3, Line 129-130
The subjects of this study were athletes from various sports such as track and field and soccer, but did you consider the or the level of competition? The level of competition will affect the frequency and intensity of each athlete's training. In addition, I should consider whether the intensity of the warm-up was appropriate for each athlete.
Page 3, lines 129-130, page 11, 397-399
The subject of this study is athletes in sports, but at Discussion it was stated that this is the first study on healthy individuals. As a basic study, what is the reason for targeting athletes with exercise habits rather than healthy adults?
Page3, Line 139-141
Please describe what was the main outcome in the previous study and how this effect size was obtained. With only the current information, I cannot determine if using an effect size of 0.3 is appropriate.
Also, I was unable to reproduce the calculation, so please provide the detailed settings of G*power.
Page3, Line 127
In measurements involving muscle output that deal with surface electromyography, it is often considered that mixing men and women in the analysis is a prudent and undesirable procedure. Please tell me if there is any harm in mixing the sexes in this study.
Page3, Line 146-147
Please explain why 80W and 100W are appropriate. The amount of load for the warm-up differed for men and women, but if the amount of load was to be changed for each person, it was necessary to set the load more strictly in consideration of body size and other factors.
Page 4, 153-154, page 5, 463-466.
We understood that this study investigated the effects of warm-up and stretching with some complexity-based methods applied to surface EMG signals. The protocol indicates that stretching is performed after warm-up and the STRETCH condition, but if the three conditions are to be compared, shouldn't there be a condition in which stretching is performed without warming up?
Page 4, 155-173.
Why did you set up two different stretching methods? Please describe the reasons.
Page 5, lines 195-196.
Regarding the position where the myoelectric is affixed, it was mentioned that it is the muscle belly and that the distance between electrodes is 2 cm, but I believe that there are characteristics of muscle morphology depending on the gender and the individual. Please let me know if there are any specific regulations, such as what percentage of the distal portion was used.
Page8, Line 303
Since it seems that the tendency of RMS is differ depending on muscle, please explain the cause of this difference.
Page10, Line 363- Page11, Line 377
I thought the usefulness of complexity analysis and the difference from linear analysis were new to this study, however, this paragraph discusses the effects of warm-up and static stretching, which seems to miss the point. Please clarify what you want to show in this paragraph.
Page12, Line 441
In this paragraph, you should describe the benefits of this study, such as which complexity analysis is appropriate to use in which situations, as well as future prospects.
Author Response
General comment
Thank you for submitting your manuscript. Your study is interesting in using new analytical methods. However, many methodological problems cannot be ignored. In addition, the explanation of clinical significance about the adopted analysis method is not enough. We hope that the following points will be taken into consideration and that this research will be further developed.
Response: Authors would like to thank the reviewer for the comments made for this manuscript. The reviewer will find below a point-by-point response to all comments. Modified sections within the manuscript are presented in red font.
Title
From this title, the purpose and design of the study could not be inferred. I think it would be better to use a different expression such as "Investigating the usefulness (or sensitivity) of complexity analysis...” to better link to the purpose of this study. It should also state in the title that this is a crossover design.
Response: Title has been modified according to reviewer's comment: "Usefulness of Surface Electromyography Complexity Analyses to Assess the Effects of Warm-up and Stretching during Maximal and Sub-maximal Hamstring Contractions: A cross-over, randomized, single-blind trial"
Page3, Line 105
I have difficulty understanding from the text that follows that this was a crossover design. It is not clear whether the data were measured three times, including familiarization, or whether the data were measured only during the two-day experimental session. Also, you should illustrate or clearly describe what conditions are present.
Response: Authors are sorry but all information are clearly presented in Methods. As previously indicated volunteers came to the laboratory on 3 occasions (Page 3, line 111). The first session served as familiarization and, of course, no data were recording (except anthropometrics) (Page3, line 116-118). Data were collected during the two experimental sessions at three time points (Page 3, line 118-120). Conditions are clearly presented immediately after the brief presentation of the 3 time points (Page 3, line 120-121). Moreover, it is clearly indicated that the only difference between the two experimental sessions is stretching (Page 3, line 120-121) and that these two conditions were presented in a random order (Page 3, line 122). Finally, the experimental sessions are detailed in chapter 2.3 with an illustration of the experimental design. Accordingly, and to facilitate reading, only slight alterations have been conducted:
- Page 3, line 112. We have deleted the term experimental.
- Page 3, line 119. We have replaced the terms "volunteers were tested" to "data were collected"
- Page 3, line 118. We moved figure 1 to this point for clarity and figure 1 caption has been modified.
Page 3, lines 120-121.
The subject's leg was listed as the right leg, but I think you need to state why it is the right leg. Also, the subject's dominant leg was not stated, and I think it is important to state the dominant and non-dominant leg.
Response: In the present study, limb dominance was ignored, and the right side was arbitrarily chosen. Ignoring the limb dominance was also more practical for the study setup. The limb dominance was not questioned. Accordingly, we can not indicate the number of individuals tested on their dominant or non-dominant limb. We modified the text and clearly indicated that the limb dominance was ignored (Page 3, L123)
Page3, Line 129-130
The subjects of this study were athletes from various sports such as track and field and soccer, but did you consider the or the level of competition? The level of competition will affect the frequency and intensity of each athlete's training. In addition, I should consider whether the intensity of the warm-up was appropriate for each athlete.
Response: We agree with the reviewer that the level of competition and usual intensity of warm-up (or more generally type of warm-up) would affect the result of the present study. The sample size used here was not large enough to make some sensitivity analyses. This limitation has been acknowledged in discussion (Page 12, line 464-467).
Page 3, lines 129-130, page 11, 397-399
The subject of this study is athletes in sports, but at Discussion it was stated that this is the first study on healthy individuals. As a basic study, what is the reason for targeting athletes with exercise habits rather than healthy adults?
Response: This inconsistency has been corrected (Page 11, Line 402). Healthy has been replaced by athletes (as indicated in the study main aim in introduction).
As indicated by the reviewer, athletes have exercise habits that could influence our results. In contrast, healthy individuals have not. However, they are not used to the different exercises used and not used to produce maximal isometric contractions of hamstring muscles. These two possibilities have different pros and cons. However, because, the study primarily referred to exercise, considering athletes seemed more appropriate.
Page3, Line 139-141
Please describe what was the main outcome in the previous study and how this effect size was obtained. With only the current information, I cannot determine if using an effect size of 0.3 is appropriate.
Also, I was unable to reproduce the calculation, so please provide the detailed settings of G*power.
Response: The main outcome was the root mean square of the EMG. It has been indicated Page 4, Line 145-146. G*power was applied using an a priori ANOVA with repeated measures (3 measurements and 2 conditions). The effect size was 0.35 (authors apologize of the rounded value given in the previous version of the manuscript) and a power of 0.8. This effect size corresponded to 0.11 as partial eta square which corresponds to a medium effect size. Sample size was 16. For clarity, the exact value of the effect size has been corrected (not rounded value) (see Page 4, line 145).
Page3, Line 127
In measurements involving muscle output that deal with surface electromyography, it is often considered that mixing men and women in the analysis is a prudent and undesirable procedure. Please tell me if there is any harm in mixing the sexes in this study.
Response: Mixing sexes could impact results not only for electromyography but also for the general response to exercise. We have previously done informal statistics to explore any possible adverse sex effect. No differences were obtained between men and women.
Page3, Line 146-147
Please explain why 80W and 100W are appropriate. The amount of load for the warm-up differed for men and women, but if the amount of load was to be changed for each person, it was necessary to set the load more strictly in consideration of body size and other factors.
Response: Of course, individualizing the cycling power output during warm-up would have been more appropriate. However, applying different intensities with gender (e.g., Su et al. J Sport Rehab 2017, 26(6), 469-477) or applying similar intensities are commonly used in the literature (e.g., Faulkner et al. Med Sci Sports Exerc, 2013, 45(2), 359-365).
Page 4, 153-154, page 5, 463-466.
We understood that this study investigated the effects of warm-up and stretching with some complexity-based methods applied to surface EMG signals. The protocol indicates that stretching is performed after warm-up and the STRETCH condition, but if the three conditions are to be compared, shouldn't there be a condition in which stretching is performed without warming up?
Response: The two experimental conditions included a standardized warm-up followed but two different stretching procedure. The aim of the present study is not to explore the specific effects of stretching without warm-up since although not recommended athletes often performed stretching during warm-ups. For clarity some sentences of the manuscript have been altered to clarify that the present study primarily investigated warm-up that included some stretching exercises (aim in abstract (Page 1, line 29-30), aim in introduction (Page 3, line 101-102) and first sentence in conclusion (Page 12, line 469-470)).
Page 4, 155-173.
Why did you set up two different stretching methods? Please describe the reasons.
Response: Static stretching is often investigated in the literature. In contrast, the other modalities are less investigated and very few studies considered neurodynamic stretching techniques. Therefore, it is of interest to determine whether these neurodynamic techniques could be recommended as part of a warm-up routines (in replacement of static stretching; not recommended during a warm-up routine). The introduction has slightly been modified to indicate that these neurodynamic techniques should be explored when included inside a warm-up routine. Please see Page 2, line 65 and line 75-76.
Page 5, lines 195-196.
Regarding the position where the myoelectric is affixed, it was mentioned that it is the muscle belly and that the distance between electrodes is 2 cm, but I believe that there are characteristics of muscle morphology depending on the gender and the individual. Please let me know if there are any specific regulations, such as what percentage of the distal portion was used.
Response: Electrodes were positioned according to SENIAM recommendations (depending on individuals' morphology). It has been added on Page 5, Line 199-200.
Page8, Line 303
Since it seems that the tendency of RMS is differ depending on muscle, please explain the cause of this difference.
Response: We agree with the fact that some RMS differences were observed for ST but not for BF muscle. As indicated in results, ST RMS was greater during the static condition as compared to neurodynamic. Such result is neither related to warm-up nor to stretching but likely to some methodological aspects (impedance or electrodes position; already acknowledged in discussion). When considering the time effect, although a significant main ANOVA effect, no pairwise differences were obtained following the post-hoc analyses. Moreover, statistics performed here were not designed to compare muscles. Accordingly, we cannot discuss any potential differences between muscles since it was not tested.
Page10, Line 363- Page11, Line 377
I thought the usefulness of complexity analysis and the difference from linear analysis were new to this study, however, this paragraph discusses the effects of warm-up and static stretching, which seems to miss the point. Please clarify what you want to show in this paragraph.
Response: The first two paragraphs of the discussion were only related to the commonly used analyses (torque and RMS). It is very important to discuss these outcomes to make clear that our results are in accordance with previously published results. The following paragraphs were related to the complexity-based approaches. Because we believe it is mandatory to have this organization of the discussion, no modification has been made.
Page12, Line 441
In this paragraph, you should describe the benefits of this study, such as which complexity analysis is appropriate to use in which situations, as well as future prospects.
Response: We believe this paragraph already clearly respond to the reviewer comments. On Page 11, Line 445-447, it is clearly indicated that entropy and DFA are better than RQA. Moreover, on page 11, line 462, we clearly indicate that other complexity-based approaches (such as multifractal DFA or multiscale entropy) should be explored.
Reviewer 2 Report
The manuscript is the first that explored the effects of warm-up and stretching with some complexity-based methods (SampEn, RQA, DFA) applied to the surface EMG signal. It is concluded that these methods are sensitive to detect some neuromuscular alterations (increase in complexity) following a comprehensive and realistic warm-up. The complexity increase obtained following warm-up suggested an optimized neuromuscular system that might be more efficient for subsequent physical activities. Non-linear complexity-based analyses therefore appeared efficient to give additional information than linear EMG analyses. The research design is well organized as well as the statistics and analysis of the results. This study was a single-blind, randomized, crossover study. Accept in present form.
Author Response
Authors would like to thank the reviewer for the expertise and for the positive comments made.Reviewer 3 Report
I reviewed the manuscript entitled “Complexity Analyses of Surface Electromyography to Assess the Effects of Warm-up and Stretching during Maximal and Sub-Maximal Hamstring Contractions"
This manuscript represents an exploratory study aimed to apply different complexity-based methods to surface electromyography (EMG) of hamstring muscles to detect changes in neural activation after a standardized warm-up and after stretching exercises. This study provides additional knowledge on the sensitivity of EMG signal for neuromuscular fatigue.
I have the following minor comments to support the improvement of the quality of the manuscript:
Introduction
(Line 47-48): "Warm-up includes various exercises at low and high-intensity aiming to improve performance and prevent injuries" Please consider the following reference: Patti A, et al. Effects of 5-Week of FIFA 11+ Warm-Up Program on Explosive Strength, Speed, and Perception of Physical Exertion in Elite Female Futsal Athletes. Sports. 2022; 10(7):100.
Materials and Methods
Please define the number of athletes for each type of sport (track and field, handball and soccer).
Please report the type of sampling.
The authors stated that they set the power at 80W for women and 100W for men. Could the authors specify how they established these powers?
Did the authors also monitor for the pedaling cadence? If so, could they specify the rotations per minute?
Authors stated that the tests tests were conducted for the right knee flexors. I was wondering if the authors have recorded lower limb dominance.
Discussion
Please specify whether the hypothesis formulated has been confirmed or not.
Author Response
I reviewed the manuscript entitled “Complexity Analyses of Surface Electromyography to Assess the Effects of Warm-up and Stretching during Maximal and Sub-Maximal Hamstring Contractions"
This manuscript represents an exploratory study aimed to apply different complexity-based methods to surface electromyography (EMG) of hamstring muscles to detect changes in neural activation after a standardized warm-up and after stretching exercises. This study provides additional knowledge on the sensitivity of EMG signal for neuromuscular fatigue.
I have the following minor comments to support the improvement of the quality of the manuscript
Response: Authors would like to thank the reviewer for the comments made for this manuscript. The reviewer will find below a point-by-point response to all comments. Modified sections within the manuscript are presented in red font.
Introduction
(Line 47-48): "Warm-up includes various exercises at low and high-intensity aiming to improve performance and prevent injuries" Please consider the following reference: Patti A, et al. Effects of 5-Week of FIFA 11+ Warm-Up Program on Explosive Strength, Speed, and Perception of Physical Exertion in Elite Female Futsal Athletes. Sports. 2022; 10(7):100.
Response: Authors are sorry but this reference was not added for several reasons. First, although this study is related to warm-up, it considered long term effects (5 weeks) while the present study considered acute effects. Second, it does not give supplementary information. Third, references cited in this sentence are more historical than the reference suggested by the reviewer and the scientific literature often forget such pioneering papers. This reference has therefore not been added.
Materials and Methods
Please define the number of athletes for each type of sport (track and field, handball and soccer).
Response: This information has been added on Page 4, Line 135-136.
Please report the type of sampling.
Response: Authors are not sure to understand this comment. The sample included competitive athletes and all were recruited from the sport science faculty in Dijon (France). This information has been added on Page 4, Line 134-135.
The authors stated that they set the power at 80W for women and 100W for men. Could the authors specify how they established these powers? Did the authors also monitor for the pedaling cadence? If so, could they specify the rotations per minute?
Response: Applying different intensities with gender (e.g., Su et al. J Sport Rehab 2017, 26(6), 469-477) or applying similar intensities are commonly used in the literature (e.g., Faulkner et al. Med Sci Sports Exerc, 2013, 45(2), 359-365). Accordingly, we determined these intensities according to multiple previously published manuscripts. A constant power ergocycle was used. Accordingly, the resistance was directly adjusted depending on the pedaling cadence. However, we did not register any information from this part of the warm-up and cannot add this information in the revised version of the manuscript.
Authors stated that the tests were conducted for the right knee flexors. I was wondering if the authors have recorded lower limb dominance.
Response: In the present study, limb dominance was ignored, and the right side was arbitrarily chosen. Ignoring the limb dominance was also more practical for the study setup. The limb dominance was not questioned. Accordingly, we can not indicate the number of individuals tested on their dominant or non-dominant limb. We modified the text and clearly indicated that the limb dominance was ignored (Page 3, L123)
Discussion
Please specify whether the hypothesis formulated has been confirmed or not.
Response: The fact that our hypothesis is confirmed is already present in discussion on Page 10, Line 362-363.
Round 2
Reviewer 1 Report
General comment
Thank you for your revise. Some of the answers to the points are still inadequate. Please respond as follows comments.
Page3,Lines 108
Although I understood why you focused the neurodynamic stretching, it is not clearly why you conduct it in this study design. Did you compare the stretching effect separately? It is problematic to lump them together as a STRETCH.
Page3,Line127
You state that you have done informal statistics to explore the possibility of adverse effects of gender differences and have obtained no differences between the sexes. I think it would be clear that there is no problem by disclosing the data and detailed statistical methods used to do so.
Page5,Lines 196
You have not separated the results by gender, but I believe the values for torque, for example, would vary significantly by gender. Is there any kind of normalization by weight or something like that?
Figure1
From your revision, it changes easy to understand that this study is used the crossover design. However, figure1 is not enough. Since figure is seen by readers at first, I recommend modifying it. Please revise to be visually recognizable that what this study design and when you assess the EMG.
Author Response
Thank you for your revise. Some of the answers to the points are still inadequate. Please respond as follows comments.
Response: Authors would like to thank the reviewer for the comments made for this manuscript. The reviewer will find below a point-by-point response to the 3 comments. Some slight alterations have been made in Methods and Figure 1.
Page3,Lines 108
Although I understood why you focused the neurodynamic stretching, it is not clearly why you conduct it in this study design. Did you compare the stretching effect separately? It is problematic to lump them together as a STRETCH.
Response. 1) As indicated on page 4 line 177, tests were conducted at 3 time points: before warm-up, after warm-up and after stretching procedures. These times points are called BASELINE, WARM and STRETCH. STRETCH therefore correspond to the abbreviation or name of this time point. 2) As indicated in statistics on page 7, line 288, a 2-way ANOVA was performed (condition x time). Time (page 7, line 290-291) correspond to the 3 tests (previously cited). Condition (page 7, Line 289-290) corresponded to the 2 experimental conditions. It means that the 2 different stretching procedures are not tested together. They are tested separately. 3) However, as indicated in Table 1, the condition effect was only observed for the ST RMS and no interactions (condition x time) were obtained. Because the statistics do not indicate any difference between the 2 experimental conditions, the discussion globalize STRETCH and do not differentiate static and neurodynamic stretching exercises in results section (not in figures).
To clarify that STRETCH is just a name, we slightly modified page 4, line 177-179.
Page3,Line127 and Page5,Lines 196
You state that you have done informal statistics to explore the possibility of adverse effects of gender differences and have obtained no differences between the sexes. I think it would be clear that there is no problem by disclosing the data and detailed statistical methods used to do so.
You have not separated the results by gender, but I believe the values for torque, for example, would vary significantly by gender. Is there any kind of normalization by weight or something like that?
Response. Because the 2 comments are related to a potential gender effect, authors have merged both comments and will answer the 2 concomitantly. 1) Of course torque output is different depending on the gender. In the previous round, authors indicated no difference was obtained between men and women. These informal statistics only corresponded to the main aim of study: effect of the experimental conditions. A three-way ANOVA was conducted. Of course, we obtained a main gender effect for torque (not for EMG analyses). However, while considering a possible interaction between gender and time and/or condition (gender x time, gender x condition and gender x time x condition) no difference was observed. 2) The aim of the study is not to test a gender effect and the design was not built to investigate this effect. If so, the number of included men and women would have been similar and larger in order to obtain high statistical power. 3) Authors used the term informal in the previous round because, the number of men and women volunteers was different and small for women. Accordingly, statistics were conducted but, because of this small number, the statistical power was low. 4) Numerous studies included individuals from both sexes without separately analyzing them (e.g., Young and Behm, J Sport Med Phys Fitness 2003 or Parikh et al., Medicine 2022).
In conclusion, because the study is not designed to investigate the gender effect, because it does not add useful information and because numerous previously published paper included volunteers from both gender without trying to differentiate them (in statistical analyses), authors have not modified the manuscript and have not included the results
Figure1
From your revision, it changes easy to understand that this study is used the crossover design. However, figure1 is not enough. Since figure is seen by readers at first, I recommend modifying it. Please revise to be visually recognizable that what this study design and when you assess the EMG.
Response. Figure 1 initially aimed to illustrate the two experimental sessions and not the general study design. However, according to your comment, this figure has been modified to indicate that conditions were randomized and that a cross-over design has been applied. EMG has also been added.